# Improving Access to Diagnostics for Schistosomiasis Case Management in Oyo State, Nigeria: Barriers and Opportunities

**DOI:** 10.3390/diagnostics10050328

**Published:** 2020-05-20

**Authors:** G-Young Van, Adeola Onasanya, Jo van Engelen, Oladimeji Oladepo, Jan Carel Diehl

**Affiliations:** 1Sustainable Design Engineering, Delft University of Technology, 2628CE Delft, The Netherlands; g.y.van@tudelft.nl (G.-Y.V.); A.A.Onasanya@tudelft.nl (A.O.); J.M.L.vanEngelen@tudelft.nl (J.v.E.); 2Department of Economics and Business, University of Groningen, 9747AE Groningen, The Netherlands; 3Department of Health Promotion and Education, University of Ibadan, 200212 Ibadan, Nigeria; oladepod@yahoo.com

**Keywords:** schistosomiasis, barriers to diagnostics, access to healthcare, end-user perspectives, neglected tropical diseases, Nigeria, case management

## Abstract

Schistosomiasis is one of the Neglected Tropical Diseases that affects over 200 million people worldwide, of which 29 million people in Nigeria. The principal strategy for schistosomiasis in Nigeria is a control and elimination program which comprises a school-based Mass Drug Administration (MDA) with limitations of high re-infection rates and the exclusion of high-risk populations. The World Health Organization (WHO) recommends guided case management of schistosomiasis (diagnostic tests or symptom-based detection plus treatment) at the Primary Health Care (PHC) level to ensure more comprehensive morbidity control. However, these require experienced personnel with sufficient knowledge of symptoms and functioning laboratory equipment. Little is known about where, by whom and how diagnosis is performed at health facilities within the case management of schistosomiasis in Nigeria. Furthermore, there is a paucity of information on patients’ health-seeking behaviour from the onset of disease symptoms until a cure is obtained. In this study, we describe both perspectives in Oyo state, Nigeria and address the barriers using adapted health-seeking stages and access framework. The opportunities for improving case management were identified, such as a prevalence study of high-risk groups, community education and screening, enhancing diagnostic capacity at the PHC through point-of-care diagnostics and strengthening the capability of health workers.

## 1. Introduction

Schistosomiasis is a parasitic disease that affects over 200 million people around the world, and 90% of the infected population are in African countries. These countries have the highest burden of morbidity and mortality [1]. Nigeria is the most endemic country in Sub-Saharan Africa with 101.3 million people at risk, and 29 million infected [2]. The infection can cause anaemia, growth stunting, cognitive impairment, decreased productivity and long-term health consequences such as bladder cancer and infertility [3]. Despite its high socio-economic burden [4], it has received limited attention from governments and stakeholders in healthcare settings, similarly to other Neglected Tropical Disease [2]. Although a prevalence study for schistosomiasis in Nigeria was conducted in 2015 [5], the selection of the sample collection was limited to children and did not address other high-risk groups such as adults [6,7].

Currently, vertical and horizontal programs are used for schistosomiasis control in Nigeria [8]. The control and elimination program is a vertical approach and a principal strategy for control of schistosomiasis (and other NTDs). The horizontal approach is the case management of individual cases at the primary health care level [9]. The control and elimination program provides annual mass treatment of praziquantel for school-age children aged 5 to 14, who are known as the most heavily infected part of the population [10]. Praziquantel has been reported to be a safe and effective treatment, and this approach is said to significantly reduce the prevalence of schistosomiasis and the intensity of infection in high endemic areas [11]. However, three major limitations characterised this approach including high re-infection rates [12], unsustainable mapping and delivery with its high dependency on donations of praziquantel [13,14], and exclusion of other high-risk groups such as people who frequently have contact with water for domestic and professional purposes [2,11].

In light of these limitations, there is a need to pay more attention to the horizontal approach (case management) because it can provide more sustainable, efficient and more localized interventions [9]. The case management approach, which is strongly recommended by the WHO, focuses on diagnosis and treatment [15,16]. In the event that the health facility does not have the diagnostic capability, symptom-based case detection is recommended. This approach has strong potentials in reducing disease transmission by shortening the infectious period of patients through early diagnosis and immediate treatment which will result in improved treatment outcomes [16].

The standard method for schistosomiasis diagnosis is microscopic examination in a lab-setting. The samples for *Schistosoma haematobium* (*S. haematobium*) are prepared either by urine filtration (using polycarbonate filters) or centrifugation. The samples for *Schistosoma mansoni (S. mansoni*) are prepared by Kato Katz faecal smear [17,18]. The challenges for sample preparation within sub-Saharan African context include the shortage of lab technicians and equipment at primary health care level [19] as well as the high labour-intensiveness and initial and maintenance costs [18]. There are alternative diagnostics methods, however, they have limitations [17]. Methods such as questionnaires, visible haematuria and urine reagent strips are available but have low sensitivity and specificity. Antibody or antigen detection-based tests are not yet commercially available. Point-of-care circulating cathodic antigen (CCA) test is on the market with high sensitivity and specificity, yet it is more specific to *S. mansoni* and has a disadvantage in affordability. For the health facilities without diagnostic capability, the WHO suggests the symptom-based case detection and treatment [15,16]. This is, for example currently being used in Ghana where the healthcare workers relate blood in urine (hematuria, dysuria) to *S. haematobium* and blood in stool and abdominal discomfort to *S. mansoni* [20]. Although the symptom-based case detection seems to be an effective method for morbidity control in high endemic areas with low resources, the detection depends on the knowledge of the health workers and prior-experience with schistosomiasis patients. There is a high possibility of failing to suspect cases with non-distinct symptoms [20,21]. It is also not clear if praziquantel is available at all levels of the healthcare system to treat the confirmed cases.

Overall, having an adequate diagnostic capability is essential to proper case management, but this requires skilled personnel with sufficient knowledge and functioning equipment. There have been reports indicating poor availability of basic equipment at the primary health care facilities in Nigeria and questions have also been raised about the quality of service delivery [22,23]. This can affect the diagnostic capability within the context of case management of schistosomiasis control. Nonetheless, to our knowledge, there is no specific study that has explored this aspect critically.

Apart from the diagnostic capabilities within the healthcare system, the disease awareness and knowledge of patients can affect health-seeking behaviour. Case management works with passive case detection, which is usually triggered by patients taking action to seek care based on a number of factors. A study in Kano state in Nigeria [24] indicates that the majority of the study participants did not have knowledge on cause, signs, and symptoms of schistosomiasis, even though the majority of them indicated that they are aware of the disease. In addition, only 35% indicated that they would seek treatment from clinics and hospitals. Another study in Adamawa state in Nigeria [25] showed that around 40% of its study participants did not seek any care, 30% visited the patent medicine vendor, while only 17% went to the hospitals. It is of note that patients, when seeking care, have a high preference toward self-medication or use of traditional healers, which may be due to the poverty and physical inaccessibility [24,25]. Nevertheless, there are information gaps on whether and how the patients become aware of the early signs after getting infected, and what barriers prevent them from taking action to seek care.

Therefore, the objective of this research is to explore how the case management currently takes place in Nigeria and to identify the barriers to access from patients and healthcare workers perspective by using empirical data. This would assist us in making appropriate recommendations for future improvement on the case management.

## 2. Materials and Methods

This study was conducted as part of the interdisciplinary research project “INSPiRED”—Inclusive diagnoStics For Poverty RElated parasitic Diseases in Nigeria and Gabon funded by NWO—WOTRO Science for Global Development programme. The INSPiRED project aims to design and deliver new technical interventions for diagnostics of malaria, schistosomiasis and hookworm infection in close co-creation with local stakeholders.

### 2.1. Ethics

The study protocol was approved by the UI/UCH Joint Ethical Review Committee of University of Ibadan (10 Dec 2019) and with registration number NHREC/05/01/2008a. Study participants were provided with an information sheet explaining the objectives of the study, and all participants signed or verbally agreed to informed consent forms prior to participation.

### 2.2. Study Setting

This study took place in Oyo State, one of the 36 states in Nigeria, with an estimated population of 7.8 million people [26]. Data for this study were collected in December 2019 from two Local Government Areas (LGAs) of Oyo State; Ibadan North and Akinyele which are based in urban and rural areas respectively. The selection was based on their moderate-to-high prevalence of schistosomiasis and accessibility to the interviewees.

### 2.3. Study Sample

The study sample consisted of five categories of stakeholder based on a literature review and expert suggestions (See Table 1). All 29 respondents were purposively selected. They were contacted and informed about the research by the local research coordinator prior to the study.

### 2.4. Data Collection

We used a qualitative approach to data collection. Semi-structured interview guides with open-ended questions were developed based on the case management steps of schistosomiasis [15] and the health-seeking pathway in low-resource contexts [27] (See Figure 1). Van der Werf [15] describes the steps in passive case detection of schistosomiasis from a health care system perspective. She distinguishes five steps in the passive case detection as a liner process of infection, pathology, disease, health care visiting, and treatment. From practice and the literature, we are aware that the trajectory is more complex and can have multiple pathways within and outside the formal healthcare system. For example, informal health care providers such as Patent Medicine Vendors (PMVs) and traditional healers are known to be frequently the first choice of health-seeking by communities in Nigeria [28,29]. For this reason, we searched for a complementary model that would represent the complexity and alternative routes of healthcare-seeking behaviour of patients in Sub-Saharan Africa. This led to the work of R. E. Kohler et al. [27] who developed a six-stage health-seeking pathway based on interviews with women from Malawi. Even though their research was related to the early detection of breast cancer, it describes the complex trajectories of patients within the African context. We constructed a health-seeking pathway with six stages that was used to derive the main themes to be addressed. The questions were formulated to cover all the themes and to guide the semi-structured interviews (See Figure 1).

The study adopted a descriptive exploratory design using qualitative methods [Key informant Interview (KII) and In-depth Interview (IDI)]. The KIIs were conducted to explore the perspectives of the patients and health workers in accessing and performing the diagnosis and the case management, and IDIs to gain a broader understanding of challenges and collect insights for opportunities. The interviews were conducted in English or Yoruba (local language) for 45 min.

### 2.5. Data Entry and Analysis

All interviews were recorded and transcribed verbatim. Interviews conducted in Yoruba were translated into English by an external translator and reviewed by a language knowledge expert to ensure that the original meaning was not lost. Transcripts were analysed using the software Atlas ti version 8.4.4.

The framework of theory of access to healthcare was used to structure the data analysis and identify the barriers to access the case management of schistosomiasis. The 5A framework by Penchansky and Thomas’s [30], which includes affordability, availability, accessibility, adequacy (or accommodation) and acceptability, is commonly used. However, we adopted Saurman’s 6A Framework in our analysis because of an additional dimension on awareness considering its importance in remote and rural areas [31].

The lead researcher used pre-defined themes based on the 6A Framework on access to healthcare [31] to assign the codes using a deductive approach. The six analysed dimensions, described by their definitions and components adopted from another study [32] are shown in Table 2. Additionally, new themes were identified by inductive coding. The initial coding was done by the lead researcher and later reviewed by two other researchers of the team. This resulted in a list of barriers to access to schistosomiasis case management. Next, the themes and barriers were discussed and grouped into the six health seeking stages. Finally, the barriers were described based on (a) the six stages of the health seeking pathway and (b) in Figure 2; by the 6As of the health access framework using the health seeker or provider perspectives.

## 3. Results

### 3.1. Health-Seeking Stages Identified in the Case Management

Based on the interviews, the six stages of the patient health-seeking pathway within the case management (See Figure 1) are described by common health-seeking behaviours. Next, identified themes within each stage which act as barriers are mentioned. The overview of the themes within 6A framework can be found in the table in Appendix A.

#### 3.1.1. Stage 1: Notice and Interpret the Initial Symptoms

All the six ‘category 1′ interviewees mention the notification of blood in urine as an initial symptom for recognition of illness. Discomfort when urinating and fever were mentioned as well.


*“I told her mum to keep her eyes on him, and she later saw him urinate and sighted blood in his urine, …”.*
*—Parent of a child who had schistosomiasis from Rural LGA*

In most cases described, identification of symptoms drew the immediate attention of the community members. However, lack of knowledge and other associations connected to the symptoms became barriers for community members to seek for health support (for schistosomiasis).

• Theme 1: Lack of general knowledge on health and schistosomiasis among community.

It was often mentioned that the general understanding within the community related to healthcare is low, which leads to less active attitudes in seeking care. More specifically, the community’s knowledge of schistosomiasis related to the cause, signs, and symptoms of the disease were limited. This lack of knowledge makes it difficult for the community to interpret the symptoms, even though they noticed the initial signs.

• Theme 2: Cultural association and belief related to the symptoms

The symptoms are associated with cultural identity or beliefs within the community. This affects people not to seek for help and choose for traditional medicines in stage 4.


*“I said it’s like a cultural thing, once you have haematuria, that normalises you as a true son of the soil…”*
*—Public Health Researcher*


*“You know that dogs have blood in their urine. In the Southwest of Nigeria, it is called Atosi Aja. This is why they believe that it is not a medical condition and they prefer treating it traditionally”*
*—PHC Coordinator (Rural LGA)*


*“Some people are aware of schistosomiasis, but most people believe that the spiritual forces have cursed the victim”*
*—Community mobilizer (Urban LGA)*

#### 3.1.2. Stage 2: Monitor Changes and Infer Illness

After people become aware of the symptoms, they have the tendency to wait for several days before they take action. They monitor the symptoms and wait for the conditions to improve. It was mentioned by the guardians and parents of the patients that the symptoms are frequently associated with other diseases, for example, Sexually Transmitted Diseases (STDs). This makes people reluctant to seek care.


*“after five days…, okay you want to see if his condition will be better before deciding to take him to the hospital.”
*
*—Parent of a child who had schistosomiasis (Rural LGA)*

• Theme 3: Trying out self-medication without prescription

While monitoring the symptoms, it is common to make use of over-the-counter medications such as paracetamol or make traditional medicine at home. If the symptoms seem to disappear, people do not further seek for help. This reduces the chance of receiving appropriate care.


*“We gave him paracetamol and yet there was no difference, he was sweating, and we took him outside to take fresh air…”*
*—Parent of a child who had schistosomiasis (Rural LGA)*

#### 3.1.3. Stage 3: Decide to Seek Help

Only after the symptom persists or becomes more severe do people decide to seek help. The patients consulted with their close ones in the community about their symptoms and where to seek help.


*“Where I (people community) will go next is dependent on that. For instance, if I speak to a friend who is a pastor and he asks me to come to his church for healing prayers, then I would go to the church. If someone says that they had once experienced such and they saw a Community Health Extension Workers (CHEW), I would follow suit.”*
*—Doctor PHC (Urban LGA)*

According to the level of knowledge in healthcare and socioeconomic status, the patients choose where to seek help in the next stage. In the community, people also preferred seeking help from the informal healthcare providers such as traditional healers and PMVs considering the accessibility and affordability (Stage 4). Depending on the relationship with health workers, sometimes they reach out the formal healthcare directly (Stage 5).

• Theme 4: The symptoms are associated with STD, which causes hesitation in sharing with others.

Due to the stigma around the STD, this type of misinterpretation causes unnecessary fear and confusion.


*“…it is possible that it is a sexually transmitted disease…So it is possible that people may contract the disease but may be too shy or lack the courage to tell someone because of losing their dignity and privacy.”*
*—Guardian of child who had schistosomiasis (Urban LGA)*

#### 3.1.4. Stage 4: Seek Help from the Social Network and Informal Healthcare Providers

##### Stage 4.1. Seeking Help from Social Network

It is common to start seeking help by consulting other community members and ask for advice from someone who had similar experiences. Patients discuss with a trusted person such as family, friends, or other community members.


*“He said it just found out he urinated blood …. so when he mentioned it was where his apprentice told him there is someone that treated him when he contracted the same disease…”*
*—Traditional healer (Urban LGA)*


*“…he confided in someone that he had contracted the disease and I got to know through that person though I was warned not to ask him or pretend as if I am not aware…”*
*—Guardian of a child who had schistosomiasis (Urban LGA)*

• Theme 5: Limited access to the proper information within the community due to low awareness on schistosomiasis

Since there is generally little knowledge on Schistosomiasis, it is difficult for patients to get access to the right information via their social network.


*“he has never heard of it (schistosomiasis), he only knows about reddish urine…”*
*—Guardian of a child who had schistosomiasis 2 (Urban LGA)*

##### Stage 4.2. Seeking Help through Traditional Medicine

Traditional medicine was believed as an effective solution, especially when other people had positive experiences to relieve the similar symptoms. Moreover, traditional healers are more accessible and affordable as they are easily approached from the community, and the cost of treatment is relatively low. The belief that the disease is related to spiritual power (Theme 2) also influences patients to choose the traditional healers, who are respected among the community.


*“they probably just tell them “oh it is spiritual problem” “Oh, it’s not normal, it’s something spiritual...”*
*—Public Health Researcher*


*“It depends on customs and traditions. It also depends on the condition because they may think that the disease is as a result of witchcraft and wizardry...”*
*—PHC Coordinator (Rural LGA)*

##### Stage 4.3. Seeking Over-the-Counter Medications from PMVs or Drug Vendors

Purchasing medicines from the PMVs or drug vendors were mentioned as a typical behaviour for health-seeking. Especially in rural communities, people prefer to buy over-the-counter medicines such as paracetamol and try self-medication as described in Theme 3. The patients or the guardians visit the PMVs and consult symptoms or ask for a specific medicine. For previous cases of schistosomiasis, they purchased antibiotics and paracetamol without prescription. The health workers referred to this process as trial-and-error where patients trying out the given medicines for one to three days and come back if not effective. If the conditions of the patients are too serious for the PMVs to handle, the PMVs provide a referral for the patients to visit the health centre.


*“I usually bought drugs from drug vendors that hawks…”*
*—Mother with a treated child with schistosomiasis (Rural LGA)*


*“They want immediate solutions, so they first buy herbs or patronize the PMVs.”*
*—Community Mobilizer/CHO (Rural LGA)*


*“Because of ignorance, the people go to them because they are at every nook and cranny”*
*—MOH/PHC Coordinator (Urban LGA)*

• Theme 6: Going through the process of trial-and-error medications without prescriptions at the PMVs.

Taking medicines without a prescription is not only causing a delay in receiving the proper care but also develops resistance to drugs such as antibiotics.


*“They mostly do trial and error just in a bid to make money regardless of lacking knowledge…”*
*—NTD Officer (Urban LGA)*

#### 3.1.5. Stage 5: Reach the Primary Level Healthcare

The CHEWs or health workers at primary health centre provide health-related education on common diseases and build a close relationship with the community. This relationship increases the chance of community members contacting them or health centre when they become ill. The four community health workers we interviewed reportedly had a strong relationship with community members which positively influenced the patients’ health-seeking pathway.


*“The PHC is a bit far away from their places but they still come around because of the relationship we have with them.”*
*—CHEW (Rural LGA)*

Considering the misinterpretation of the symptoms (Theme 4), the trust between the patients and a health worker is important to open up about the symptoms.


*“Based on the relationship we have with them; they can easily tell us without feeling embarrassed or shy…. They know me, and I have been with them for a long time.”*
*—Community mobilizer (Rural LGA)*

• Theme 7: Negative attitudes of the health workers may prevent people from accessing formal health care.

The negative attitudes of some of the health workers may become potential barriers to access formal health care.


*“The attitude needs to be improved so that we can be more receptive to these people”*
*—MOH/PHC Coordinator (Urban LGA)*


*“We make sure things are friendly and simplified in order to make sure they are not scared…”*
*—NTD Officer (Urban LGA)*

##### Stage 5.1. Consultation

The consultation with health workers starts from asking about the symptoms and the patient history. If schistosomiasis is suspected, possible contact with water is also asked. The time to get attended was not considered as a challenge in both LGAs.


*“When the patients are brought to the clinic, we ask about the complaints, we find out if the child bathes near wells and rivers and they say yes....”*
*—Community mobilizer (Rural LGA)*

Once the health care workers recognize the symptoms and suspect schistosomiasis, there are two actions they should take. First, provide the appropriate case management or refer the patients to another health centre or hospital where the patients can receive the care.

Secondly, report to the Local Government as schistosomiasis to be included in the Integrated Disease Surveillance Response program. This will call the attention of the Disease Surveillance Notification Officer (DSNO), and the DSNO who will initiate the surveillance protocol to collect a sample and confirm the case. For the surveillance protocol, the DSNO collects the sample and brings it to a qualified laboratory for diagnosis. However, this does not take place for unreported cases.

• Theme 8: Knowledge gap of high-risk groups of schistosomiasis among the community

The stakeholders at the community level (Categories 2, 3) often mentioned that the prevalence is low in the LGAs where the interviews took place. However, the higher-level stakeholders (Categories 4, 5) mentioned specific communities are at higher risk of schistosomiasis, for example, around the riverine areas, which is not recognized by the health workers and cause low awareness. This is a clear gap in knowledge that can lead to the cases being missed and underreported.


*“In Ibadan (city), for instance, there is a location called Dandaaru. It is around University College Hospital. People live around that community and their children go there to bath. In the process, they get infected with schistosomiasis.”*
—Public health researcher

• Theme 9: Failure in suspecting the case based on symptoms

Most health workers are familiar with blood in urine, but they may not associate the symptoms with schistosomiasis case. In addition, there may be non-specific symptoms which makes it difficult for the health workers to identify the case. This prevents the patients from receiving the appropriate care for schistosomiasis, and the surveillance program will miss the case.


*“if a patient comes with a case of blood in their urine and if the health worker does not have adequate knowledge to say that it is similar to schistosomiasis, there is no way the patient can take a step further to investigate… There may be misdiagnosis and some cases may be entirely missed. Some may have the disease and assume that it is a sexually transmitted infections…Training of the health workers to build their skills to detect schistosomiasis is very important.”*
*—MOH/PHC Coordinator (Urban LGA)*


*“I’m not sure maybe 5 or 9 of them had a microscopic (haematuria) and not the haematuria… it wasn’t like they came with symptoms.”*
*—Public health researcher*

##### Stage 5.2. Diagnosis

Once a schistosomiasis case is suspected, a diagnosis should take place to confirm the case and provide treatment. Preventing wastage of free medication from the health centres is another factor mentioned.


*“You must carry out urinalysis with at least simple microscopy. It is very important to know what you are dealing with and to rule out certain thing…”*
*—MOH/PHC (Urban LGA)*


*“Diagnosis is very important because, without it, no treatment can be made.”*
*—PHC Coordinator (Rural LGA)*

The diagnosis will be performed within the facility if a laboratory is available and functioning. We inquired the lab scientists about the standard method for schistosomiasis which was confirmed as urine analysis. It involves the collection of a urine sample, sample preparation, urine strip tests, and microscopic analysis. Urine filtration was not mentioned by any of the four labs we visited. The sample preparation was done by centrifuge as a standard procedure.


*“Now a patient comes to the laboratory and the physician has requested a urinalysis, for a urine analysis and a urine microscopy”*
*—Lab scientist (Urban LGA)*

However, none of the four health centres we visited had a functioning laboratory in place. The barriers include incomplete infrastructure and lack of equipment, unstable electricity and power supply, and lack of qualified health care professionals.

• Theme 10: Incomplete medical infrastructure to perform diagnosis

In the primary health centre, there was a lack of adequate equipment to perform the microscopy such as centrifuge, microscope, clean and controlled environment, and a stable supply of electricity. The size of the space and the environment (excessive heat, exposure to sunlight) were also mentioned as reasons why the facility was not functioning.


*“There is no machine (microscope). We do have labs, but we are limited to some tests to be carried out at the LGA level.”*
*—NTD Officer (Rural LGA)*


*“But at times, when we don’t have equipment, we call our boss and ask to either to refer the patient or if he is on his way down, if he is, he would bring the equipment needed from Moniya by his bike…”*
*—CHEW (Rural LGA)*

Unstable power supply was another factor due to which microscopy cannot function properly. As an alternative solution, the generator was mentioned, but only two of the visited labs were equipped with it. The generators were functioning at the moment, but it was also mentioned that the generators frequently break down and do not function. Maintenance of the broken devices was an additional challenge especially when they do not have a back-up device.


*“there is currently no power supply. We have an old generator and there is no money…”*
*—Lab scientist (Urban LGA)*

• Theme 11: Lack of lab scientists and technicians to perform diagnosis

The absence of lab scientists was a recurring issue at the primary level health centres. Among the four labs we visited, we interviewed at least one lab scientists or technician. In two labs attached to the primary health centre, only lab technicians were present. Even if they are available, potential insufficient training of the professionals was considered as a barrier.


*“… Then manpower should be on ground. Scientists, more scientists should be on ground so that the work won’t be too much on individuals…”*
*—Lab scientist at PHC (Urban LGA)*


*“She is a laboratory technician, not a full scientist. She is just a technician…”*
*When asked about the lab personnel at the PHC– Head of PHC (Urban LGA)*

• Theme 12: Incapability to perform the diagnosis with sufficient quality

One academic stakeholder mentioned that the lack of skills of lab scientists and technicians is one of the challenging factors to deliver diagnostic results with sufficient quality.


*“the skill of their laboratory technician is not good enough to pick that, then you might miss even if there are 100 cases in that community…”*
*—Public Health Researcher*

• Theme 13: Extra steps of movements are required for diagnosis and treatment.

• Theme 14: The costs incurred for extra steps are patient’s responsibilities.

At the health centre without diagnostic capacity, diagnosis may be requested from other facilities. In this case, it is the responsibility of the patient to reach there and bring the results back for treatment. Some patients may not continue the health-seeking pathways due to the extra costs of transportation and time incurred. It was mentioned that the costs of diagnosis and treatment are free at the health centre. However, if these are not available at the health centre, it is under patients’ own expenses to go to a private lab for diagnosis or pharmacy for treatment.


*“our people are still poor, if test is expensive they will say they will come back…. She told me she didn’t have enough money on her for the test that she had only five hundred naira…”*
*—Guardian of child who had schistosomiasis II (Urban LGA)*

##### Stage 5.3. Symptom-Based Diagnosis and Treatment

In prior cases of schistosomiasis, health workers have provided treatment based on symptoms without diagnosis. The lack of knowledge of the health workers not only caused schistosomiasis case to be missed, but also questioned the reliability of the symptom-based treatment.


*“We combine the signs and symptoms with the patient history of the patient… We treated them clinically as we did not have any laboratory to confirm it.”*
*—Community Mobilizer (Rural LGA)*

• Theme 15: The symptom-based treatments are not always reliable.

The stakeholders from category 4 and 6 mentioned that the symptom-based treatment is not always reliable. As mentioned in theme 8 and 9, the case may not be suspected at all or fall under misdiagnosis or mistreatment.


*“Even on clinical level, such a diagnosis can be missed…So, when you have this patient and you do not use your initiative to conclude that you have to conduct urinalysis with microscopy on this patient, it is possible to miss the diagnosis…”*
*—MOH/PHC Coordinator (Urban LGA)*

##### Stage 5.4. Treatment and Follow-Up

Generally, the treatment will be provided according to the results of the diagnosis. In the previous cases, the treatment was given before receiving the diagnostic results. Even when DSNO requested the diagnosis, the treatment was given without waiting for confirmation due to the delay in receiving the results.

• Theme 16: Treatment is given before the test results are available.

This happened when there was a delay in receiving the results. For the convenience of the patients, the treatment was given immediately, so they do not have to come back for the results and treatment—This relates back to the barriers in accessibility and affordability.


*“After everything, my boss told me they got drugs and that it was schistosomiasis. However, I did not see the laboratory results.”*
*—CHEW (Rural LGA)*


*“If you ask the patient to go home without giving them anything, they will not come back to you. This is why you have to reassure and give them something without the case being confirmed”*
*—PHC Coordinator (Rural LGA)*

The follow-up takes place by the health workers to check on recovery via personal contact or phone call.


*“after that they will tell them to take their drugs properly, they will also tell them to do check-up either the following or after two days…”*
*—Guardian of a child who had schistosomiasis II (Urban LGA)*

#### 3.1.6. Stage 6: Reach the Referral and Receive Appropriate Care

From the primary health care, the patients are referred to visit an advanced level of health care facility. In the areas we conducted our study, the Hospital affiliated to the University of Ibadan was often mentioned as a referral and, in most cases, patients followed the advice for schistosomiasis and other diseases. Once they reach the referral hospital, the patients received the appropriate care with diagnosis and treatment. However, there were still barriers such as the long-distance to the healthcare provider, the transportation costs and the general fear of health care.

• Theme 17: Distance to the health care provider is far.

• Theme 18: Transportation costs are unaffordable.

In the communities located in urban areas, access to the referred healthcare facility such as hospital was not described as a challenge. However, in the rural communities, the distance to the healthcare facility was major challenge in access as well as costs of transportation. Moreover, it was described that the patients might have to depend on other family members or neighbours to arrange transportation which cause additional delay in seeking care. One health worker mentioned that she offers to provide the transportation costs when she gives a referral. The time to travel to the health care provider was also seen as a challenge.


*“Even transportation is a cause for concern. They want immediate attention and asking them to go to another hospital is like adding salt to their journey…”*
*—Community mobilizer/CHO (Rural LGA)*


*“Even if free drugs are available at the hospital, they have to think of the transport fare from their house to the hospital.”*
*—PHC Coordinator (Rural LGA)*

• Theme 19: Fear of healthcare facility and uncertainty make people hesitant to reach referral

Another barrier identified was the general fear for healthcare facility as the hospital was usually associated with a place “with stress” due to their prior experiences. The fear comes from the uncertainty of the further process in which they might have to spend excessive time and costs.


*“We have heard of cases of people with phobias for health center that close their eyes when they walk pass by the facility”*
*—NTD officer (Urban LGA)*


*“I said fear and shyness, fear that they will be admitted (to hospital) and may not be allowed to come go back home…”*
*—Guardian of child who had schistosomiasis (Urban LGA)*

### 3.2. Barriers to the Case Management and Diagnosis

In the final stage, the identified barriers were grouped according to the six stages of the healthcare seeking pathway and the 6A dimensions of access to healthcare (See Figure 2). It is also indicated if the barriers were from the perspectives of healthcare seeker or a provider.

## 4. Discussion

### 4.1. Main Findings

This study explored the health-seeking behaviours of patients and the diagnostic capability of the primary healthcare level in schistosomiasis case management in Oyo State, Nigeria. Based on the results, we identified barriers to access to adequate health care and diagnosis. The overall health-seeking pathway was found to be in line with the pathway as identified in the literature [15,27]. We elaborated on the health-seeking behaviours and barriers within each stage specifically related to schistosomiasis. Overall, the barriers from the healthcare seeker perspectives were spread over all six stages (See Appendix A). The disease awareness was major barrier for the patients at the beginning of the health-seeking pathway, followed by accessibility and affordability. The barriers from the provider perspectives were more present in the later stage of pathway (Stage 5 and 6) where the availability of diagnostics and disease detection rates by the health worker were challenging factors. During the interview, no other categories of health-seeking stages and barriers to access were mentioned which indicates the validity of the framework we used.

As mentioned in previous studies [24,25], awareness was one of the main barriers. Within our empirical study, a lack of awareness was spread over several stages of the health-seeking pathway. The barriers to awareness are more present in the early stages of health-seeking before visiting the health facilities and from the patients’ perspectives. The low awareness of the disease and other associations connected to the symptoms in the community resulted in delay in seeking healthcare and self-medication. The patients often choose alternative routes (PMVs and traditional medicine) before they reach formal healthcare. This is similar to the health-seeking behaviour of other common diseases [28,29] using trial and errors of medication without prescription which causes delay in accessing care.

When patients seek out formal health care, schistosomiasis is often underestimated by the health workers. The barriers to the acceptability strongly influenced the decision-making where the health care workers attitudes and perception of formal health care were the major issues. The health care providers have multiple barriers related to accommodation including the knowledge gap and non-specific symptoms of the disease. Even when the case is suspected, the health care providers face challenges in providing laboratory confirmation of the case due to unavailability of the diagnostic equipment and personnel. The unavailability issues were directly related to the incomplete infrastructure, lack of training of personnel, and environmental challenges such as power supply. This leads to failure in following the recommended procedure of the case management and raises additional barriers to accommodation. The symptom-based treatment, which is alternative to diagnosis, was found to be frequently not reliable due to the limited knowledge of the healthcare providers. Referral to another diagnostic facility is possible but the delay in receiving the result led to treating the patients without confirmation for the sake of convenience. Lastly, the referral to other facilities brings more burden of time and costs for the patients, which relates to the issues of accessibility and affordability.

### 4.2. Opportunities

Based on the identified barriers and comparing our findings with existing knowledge, we suggest following opportunities to improve access to the proper case management and diagnosis.

#### 4.2.1. Community Sensitization Program for Awareness Creation

Active involvement of the community members for sensitization and health education will improve the general awareness on schistosomiasis by overcoming mis-associations around the symptoms and the passive attitudes in health-seeking behaviours. There is a need for focusing on the information on the disease causation, risk practices, key symptoms, consequences of the disease and delayed treatment. This will influence the community members to take desirable actions for prevention and seeking care once they notice early signs of the schistosomiasis. Multiple methods of dissemination should be used including informal healthcare providers such as traditional healers and PMVs.

#### 4.2.2. A Study to Identify Prevalence of Schistosomiasis Among other High-Risk Groups

There is a need for a prevalence study focusing on other high-risk groups including the adults who frequently interact with water. Since children are already covered by the control and elimination program, this will help the health workers to realize the hidden burden of schistosomiasis in their local context. As the prevalence of schistosomiasis is perceived as low without evidence, schistosomiasis is “neglected” among the community and health workers. Presenting data specific to their local context will provide the health workers with awareness of the severity and urgency and consequently improve accommodation by providing the appropriate care for schistosomiasis. The prevalence study will generate needed evidence and guide the development of appropriate strategies for effective implementation of case management.

#### 4.2.3. Enhancing the Existing Diagnostics Capacity

Stimulating adequate case management of schistosomiasis infection requires minimizing the number of steps by patients to reach the health centres where they can receive appropriate care. In this study, poor accessibility is evident with a concomitant effect on PHC utilisation. The issue of transportation costs due to referrals cannot be resolved without improving the accessibility and availability of the health workers and equipment. Enhancing the existing diagnostic capacity at the primary level will reduce the additional movements to reach other health facility and laboratories. Complementing the existing laboratory infrastructure, provided with more equipment and skilled personnel, will be necessary as well as providing solutions for other environmental factors such as unstable power supply.

#### 4.2.4. Implementation of Point-of-Care Diagnostics Solution

Implementing an affordable and simple point-of-care diagnostics solution will reduce the financial burden of equipment and personnel at each health facility. Point-of-diagnostics can confirm the detected cases immediately and will reduce the risk of missed or misdiagnosed cases. It will be a favourable solution to allow the task distribution with minimal training, for example, by enabling community-based diagnosis by the community health workers. If the sensitivity is sufficient enough, the point-of-care diagnostics solution can be utilized for other opportunities such as prevalence study or community-based screening with higher affordability and accessibility. From the patient perspectives, additional travel and costs to the diagnostic facility are no longer necessary and there will be no delay in the results.

#### 4.2.5. Community-Based Screening for Treatment and Monitoring

It would be a feasible approach to improve the availability of diagnostics at the primary healthcare level by adding schistosomiasis screening to other health care interventions already in place. This will likely have an immediate impact in reducing the number of infections and help in the collection of data for prevalence monitoring. A new diagnostic method available at the point-of-care with smart, high sensitivity and immediate output generation will add immense value to carry out such screening and would stimulate demand for services. This will be an advantage for the health centre in rural areas by preventing additional travel to the diagnostic facility.

#### 4.2.6. Capability Strengthening of the Health Workers

Increasing the capability of the health workers will be a key to improve detection rate at the community level. More suitable approaches for different endemic levels should be determined based on the prevalence study and guide the health workers. The training of the health workers should include suspecting schistosomiasis cases from the symptoms and the contextual factors and emphasize the importance of the diagnostics. The availability of smart diagnostics will be beneficial to detect light infections or asymptomatic cases and avoid misdiagnosis. The willingness of people to seek help in formal health care was strongly influenced by the close relationship between the health workers and the community. Accordingly, the positive attitudes of the health workers towards the community should be emphasized in the training. New interventions should consider training the health care providers at the community level and the informal sector (PMVs and traditional medicine) to enhance collaboration between them. This will improve the awareness in the community.

### 4.3. Limitations of the Study

The limitations of this study should be noted. First, the patient’s experience of schistosomiasis (stakeholder category 1) have taken place in the last 3 years. However, we were able to validate the findings and gain a more comprehensive understanding of other contextual and organizational factors through interviewing the multiple levels of stakeholders. The stories of the cases in the past were validated by confirming the facts with the health workers who put us in contact with the respondents as they were involved in the case as well. Second, the number of stakeholders were limited and they were selected from two LGAs, which can limit the generalizability of the results. Even though there was still limited access to health care and diagnostics, the LGAs were still considered to be close to the urban area (capital of the state). More studies can be conducted in more rural areas to deepen the understanding of barriers more specific to that context. Nevertheless, we believe the key findings and the identified barriers from this study are generalizable to similar settings and can be used to improve the case management of schistosomiasis.

## Figures and Tables

**Figure 1 diagnostics-10-00328-f001:**
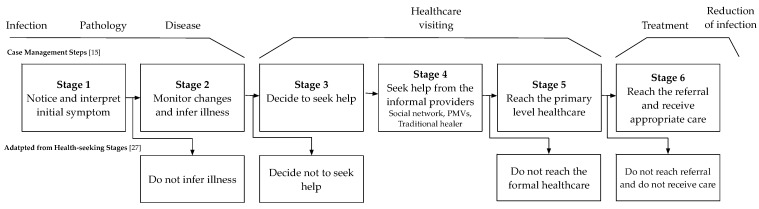
Adapted health-seeking pathway with six stages based on [15,27].

**Figure 2 diagnostics-10-00328-f002:**
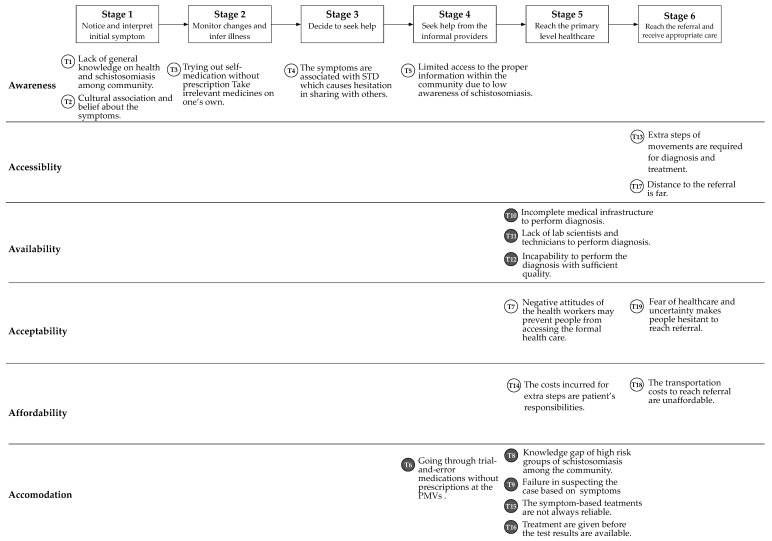
Identified barriers in the case management and diagnosis in 6A Framework. ○ = Barriers from the healthcare seeker perspectives, ● = Barriers from the healthcare provider perspectives.

**Table 1 diagnostics-10-00328-t001:** Stakeholder categories and respondents.

	Stakeholder Categories	Respondents	LGA
1	Community members who have experience with schistosomiasis	6 Parents/Guardians of people who were treated for schistosomiasis	Ibadan North, Akinyele
2	Stakeholders within community that can impact on the patient decision to access care	1 Traditional healer 1 Community leader 1 Patent Medicine Vendor (PMV)	^1^ Ibadan North
3	Stakeholders in the formal health care	2 Community Health worker 2 Community mobilizers 1 Doctor 5 Lab personnel	Ibadan North, Akinyele
4	Stakeholders within Local and State Government	1 Medical Officer of Health/PHC Coordinator 2 Disease Surveillance Notification Officers (DSNO) 1 PHC Coordinator 2 LGA NTD Officer 1 State NTD Officer	Ibadan North, Akinyele
5	Stakeholders in academia	3 Researchers	University of Ibadan

^1^ The community in Akinyele did not have residential traditional healer or PMV.

**Table 2 diagnostics-10-00328-t002:** 6A Framework of access to healthcare.

Dimensions	Component	Theme
Awareness	Communication and information	General health literacy Knowledge about symptoms, care and prevention
Accessibility	Location	Distribution of, and distance to, health care providers
Availability	Supply and demand	Incomplete medical infrastructure Lack of equipment Lack of health care professionals Lack of training for health care professionals
Acceptability	Consumer perception	Cultural belief and influence from the community
Affordability	Financial and incidental costs	Cost of treatment Cost of transport to health care provider
Adequacy (Accommodation)	Organisation	Mismatch between available information and awareness, knowledge, and education needs Lack of relevant and complete diagnostic information

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
