# Peer review of "Improving Access to Diagnostics for Schistosomiasis Case Management in Oyo State, Nigeria: Barriers and Opportunities"

_diagnostics, 2020, doi:10.3390/diagnostics10050328_

Round 1
Reviewer 1 Report
This is a very interesting, well-conducted and well-written manuscript. I have very few comments that are editorial. There are slight English corrections needed throughout but these can be picked up at the editorial stage.
Line 61 and others – all species names should be in italics.
Line 67 – state which tests you are referring too – haematuria, CCA, CAA as all differ in specificity and sensitivity.
Line 228 – can you check it is correct
Line 532 – check point-of-diagnostics
Line 555 – check the sentence as does not read well
Line 569 – change study to studies
Author Response
Thank you for your feedback. Please see below for the correction. The revised manuscript is attached as MS Word file.
Line 61 and others – all species names should be in italics.
> All the species names appear in the body have been changed to italic style.
Line 67 – state which tests you are referring too – haematuria, CCA, CAA as all differ in specificity and sensitivity.
> The specific diagnostic methods are described with brief explanations on limitation of each test.
"There are alternative diagnostics methods, however, with limitations [17]. Methods such as questionnaires, visible haematuria and urine reagent strips are available but have low sensitivity and specificity. Antibody or antigen detection-based tests are not yet commercially available. Point-of-care circulating cathodic antigen (CCA) test is on the market with high sensitivity and specificity, yet it is more specific to S. mansoni and has a disadvantage in affordability."
Line 228 – can you check it is correct
> The typos have been corrected as “…it is possible that it is a sexually transmitted disease…So it is possible that people may contract the disease...."
Line 532 – check point-of-diagnostics
> Corrected from "Point-of-diagnostic" to "Point-of-diagnostics"
Line 555 – check the sentence as does not read well
> The sentence was rewritten as "The willingness of people to seek help in formal health care was strongly influenced by the close relationship between the health workers and the community. Accordingly, the positive attitudes of the health workers towards the community should be emphasized in the training."
(Original sentence: The willingness of people to seek help in formal health care was strongly influenced by the close relationship between the health workers and the community. Accordingly, the positive attitudes of the health workers towards the community should be emphasized in the training.)
Line 569 – change study to studies
> Corrected
Reviewer 2 Report
Important paper! This is a much needed reminder that sucessful schistosomiasis control depends on numerous factors which need to be identified at the grass-root level.=
Author Response
No specific comments for revision.